# LEARNING HOW NOT TO ACT IN TEXT-BASED GAMES

**Matan Haroush, Tom Zahavy, Daniel J. Mankowitz and Shie Mannor**
The Technion - Israel Institute of Technology
{matan.h@campus,tomzahavy@campus,danielm@campus,shie@ee}.technion.ac.il

## ABSTRACT

Large actions spaces impede an agent's ability to learn, especially when many of the actions are redundant or irrelevant. This is especially prevalent in text-based domains. We present the action-elimination architecture which combines the generalization power of Deep Reinforcement Learning and the natural language capabilities of NLP architectures to eliminate unnecessary actions and solves quests in the text-based game of Zork, significantly outperforming the baseline agents.

## 1 INTRODUCTION

*"This is an open field west of a white house, with a boarded front door. There is a small mailbox here. A rubber mat saying 'Welcome to Zork!' lies by the door"*. This is an excerpt of the opening clues provided to a player in "Zork I: The Great Underground Empire"; one of the first interactive fiction computer games, created by members of the MIT Dynamic Modeling Group in the late 70s. The game immerses players in a strange reality where very little information is available. By exploring the world via interactive text-based dialogue, the players progress in the game. The world of Zork presents a rich environment with more than 200 different locations and abstract objects (see Figure 1. In this paper, we focus on quests related to the zoomed sub-section of the map).

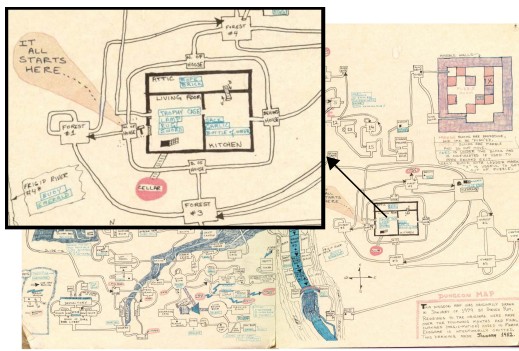

Figure 1: The world of Zork

Zork players are requested to describe their actions using language instructions. For example, in the opening excerpt, an action might be 'open the mailbox' (Figure 1). Once the player describes his/her action, it is processed by a sophisticated natural language parser. Based on the parser results, the game presents the the outcome of the player's action. A scalar reward is given to the player when a task has been completed. Zork presents multiple challenges to the player like building plans to achieve long-term goals, remembering implicit clues as well as learning the interactions between objects in the game and specific actions.

In this work, we focus on solving the game of Zork using Artificial Intelligence; in particular, using the combination of Deep Reinforcement Learning (DRL) – to generalize across states – and Natural Language Processing (NLP) – to deal with text-base state and action spaces.

The DQN (Mnih et al., 2015) is a DRL algorithm that achieved unprecedented success in solving Atari games by learning to approximate a Q function using a Deep Neural Network (DNN). This function approximation enables the DQN to generalize across states – thus, dealing with the challenge of exponentially large state spaces. In contrast to Atari's finite, discrete action space, Zork's action space is composed of all possible sequences of words from a fixed size dictionary resulting in a significantly larger action space.

This work proposes action elimination; that is, restricting the available actions in each state to a subset of the most likely ones, based on feedback from the emulator. The core assumption here is that it should be easier to learn to predict which actions are valid for each state and leverage that information for agent control, rather than learning the actual Q function for all possible state-action pairs. Our approach comprises two networks, a DQN and an Action Elimination Network (AEN),

both designed using DNN architectures suited to NLP tasks. The AEN eliminates irrelevant actions, and the DQN learns Q-values for the relevant actions. We test our method in Zork and demonstrate the agent's ability to advance in the game faster than the baseline agents by eliminating actions.

Previous studies have adapted the DQN algorithm to Text-Based Games such as LSTM-DQN (Narasimhan et al., 2015), and DRRN (He et al., 2015), but consider smaller action spaces and are orthogonal to this work. Other works have generalized over large discrete actions spaces (Dulac-Arnold et al., 2015), or have learned to act using continuous action spaces (Hausknecht & Stone, 2015; Masson et al., 2016). However, none of these works have considered action elimination to accelerate learning, nor implemented this in text-based domains.

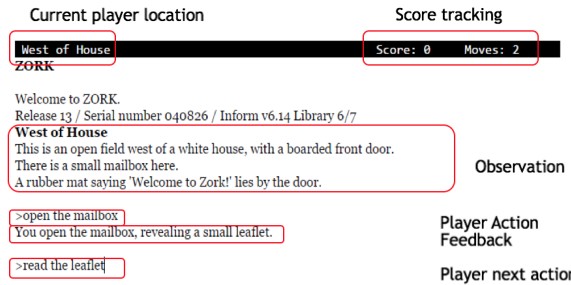

Figure 2: Zork in-game interface

## 2 METHOD

**States and Actions:** Our approach builds on the standard RL formulation (Sutton & Barto, 1998). We represent the state as a sequence of words, composed of the game descriptor (Figure 1, "Observation") and the player's inventory. These are truncated or zero-padded (for simplicity) to a length of 50 (descriptor) + 15 (inventory) words and are embedded into continuous vectors in $\mathbb{R}^{300}$ using word2vec (Mikolov et al., 2013). We consider two types of actions. Action "templates", composed of a {Verb, Object} tuple for all the objects in the game (e.g, open mailbox), and "prior knowledge" actions that are used here for practical reasons and contain simple actions that are crucial for solving the game (e.g., go east). In the next section we will show that action spaces of this size cause standard DRL approaches to struggle, emphasizing the need for action elimination.

**AEN:** The agent stores in its Experience Replay (Lin, 1992) information about states, transitions, actions, and rewards. In addition, our agent also stores feedback from the emulator regarding the validity of its actions. Based on this information, we designed an NLP CNN classification architecture, based on Kim (2014), to predict action relevance in each state. This network, refereed to as AEN is trained to minimize the BCE loss (multi-label classification) over all possible game actions, and estimates the probabilities for actions to fail in a given state.

**AE-DQN:** These predictions are then used by the Action Eliminating DQN (AE-DQN) to select actions. Our DQN architecture is tailored for text domains and uses a word2vec embedding layer followed by a simple, 2-layer NN with a ReLU activation. After embedding, the input to the network is in $\mathbb{R}^{19500}$ (embedding size times input length $300 \cdot 65$), and the hidden layer size is 100 (the size of the last layer depends on the size of the action space and is later specified per experiment).

**Action Selection:** At first, a standard epsilon-greedy mechanism is applied with linear epsilon annealing (from 1 to 0.05, similar to (Mnih et al., 2015)). If a greedy action is chosen, then the agent is using an **Action Elimination** mechanism. This mechanism restricts the agent to choose actions only from a small subset of valid actions. The subset is generated at each time step and consists of the $k$ most likely actions (sorted according to the AEN probabilities) and an additional $m$ actions that are drawn at random from a multinomial distribution w.p. softmax$(1 - \text{prediction})$ (similar to Boltzmann exploration, but on the AEN predictions). If a random action is chosen, then the agent uses the **Explore** mechanism. First, the agent selects an action $a_{rand}$ at random. If $a_{rand}$ is valid, i.e., its AEN probability is larger than a fixed threshold $\tau$, it is selected. Otherwise, with probability $1 - p_{drop}$ the agent still selects $a_{rand}$ to promote exploration or with probability $p_{drop}$, the Explore mechanism is repeated. This procedure helps the agent to select valid actions while avoiding action starvation resulting from an incorrect model.

## 3 EXPERIMENTS

**Domains:** The agent is evaluated on two quests: **(1) The Egg Quest:** The agent quest is to find and open the jewel-encrusted egg, hidden up on a tree in the forest. The agent is awarded 100 points

upon successful completion. **(2) The Troll Quest:** The agent must find a way to enter the house grab a sword and the lantern, expose the hidden entrance to the underworld and then defeat the troll guarding it, awarding him 100 points.

**Action Space:** The Zork agent is given a fixed set of actions that allow it to complete its quests; these include *navigate* (south, east etc.) *open* an item and *fight*. In addition, we augment the action space with a set of *"take"* actions for possible objects in the game. The "take" actions correspond to taking a single object, and includes objects that need to be collected to complete quests, as well as other irrelevant objects from the game dictionary.

**Setup:** The agent's goal in each quest is to maximize its cumulative reward. A reward of $-1$ is applied at every time step to encourage the agent to favor short paths. Each trajectory terminates upon completing the quest or after 100 steps are taken. For AE-DQN we chose $\tau = 0.75$ and $p_{drop} = 0.8$. Finally, we used the discounted factor $\gamma$ during training to be $\gamma = 0.8$ but use $\gamma = 1$ during evaluation (like in the DQN paper). We compared the AE-DQN agent (yellow) with the vanilla DQN (blue) as well as two ablative instances of AE-DQN. One that only uses the Action Elimination (AE-Greedy, red) and one that only uses the exploration mechanism (AE-Explore, green).

**Results:** Figure 3(a) presents the results for the Egg quest, with only 5 take actions (a total of 15 actions including the essential actions) and chose $k = 2, m = 1$ for the AE mechanism. We can see that all the agents can solve this task when the action space is small; in addition, the action elimination modifications do not harm the agent's performance. Furthermore, the learning curve shows that the modified agents can get to rewards earlier than the vanilla agent and produce a more stable policy. Figure 3(b) also presents results for the Egg quest, but now with 200 take actions (a total of 210 actions) and $k = 10, m = 5$ for the AE mechanism. We can see that increasing the number of actions makes it harder for the vanilla agents to learn an optimal policy despite the small size of the problem. On the other hand, all the action eliminating agents seem to perform better. The AE-Explore agent reaches positive reward faster but suffers from instability, while both action elimination agents are more stable.

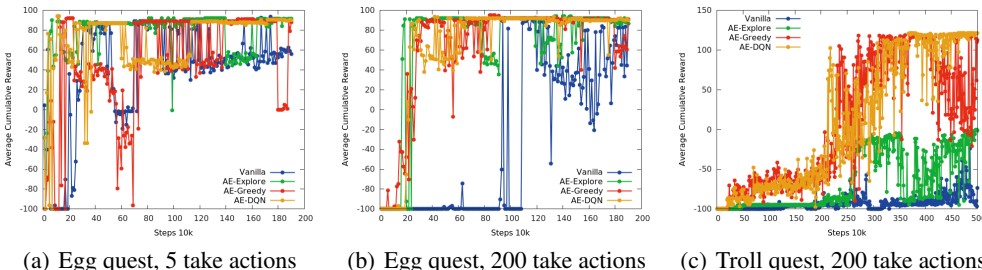

(a) Egg quest, 5 take actions     (b) Egg quest, 200 take actions     (c) Troll quest, 200 take actions

Figure 3: A comparison of agent performance in the Egg $(a),(b)$ and Troll $(c)$ quests

Finally, Figure 3(c) presents the results for the Troll quest, with 200 take actions (totaling 215 actions) with $k = 10$ and $m = 5$ for the AE mechanism. We can see that the action elimination agents solve this quest in very few iterations, outperforming the baseline vanilla DQN agent. The AE-Explore agent does not seem to perform well without the action elimination mechanism, although it performs better than the vanilla agent. Finally, we can see that both the AE-Explore and the AE-DQN agents enjoy more stable behavior as they continually explore and evaluate actions without suffering from action starvation.

## 4 SUMMARY

We proposed the AE-DQN, a DRL approach for solving text-based games that performs action elimination, effectively learning which actions **not to select**. By doing so, the size of the action space is reduced, exploration is more effective, and learning is improved. We believe that by eliminating actions, we effectively reduce over-estimation errors in state-action pairs (Hester et al., 2018). This direction has been explored less and we intend to investigate this in future work. We also plan to remove most of the prior knowledge to solve the complete Zork game with an action space that is exponential in size. This system can also potentially help in improving the performance of real-world, NLP systems that use DRL-like chat bots (Serban et al., 2017; Li et al., 2016) and personal assistants (Wu et al., 2016).

.

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
