# OpenReview forum: "Learning How Not to Act in Text-based Games"
_ICLR.cc/2018/Workshop — Accept_

### Official Review · AnonReviewer3 · 2018-03-09
**Noisy results, incremental and questionable method**

**Rating:** 4
**Confidence:** 3

**Review:**

The paper proposes an 'Action Elimination Network' (AEN) in order to impose an a-priory (state dependent) restriction on the actions considered by the RL agent. The AEN is training on a replay buffer of state, action and feedback tuples, where the feedback is coming from the emulator (indicating whether or not an action is valid).

One concern is that simply adding a shaping reward which puts a large penalty on illegal actions could in principle lead to the same learning dynamics.
Predicting whether a given action is valid by the AEN is then equivalent to predicting a large negative reward by the DQN. Obviously this would have to combined with boltzman exploration (or similar) in order to ensure that the illegal actions are selected with lower frequency.

Another concern is that the results are extremely noisy. They seem to be based on a single random seed and lack any statistical validation.

---

### Official Review · AnonReviewer2 · 2018-03-09
**Interesting take on large action spaces**

**Rating:** 8
**Confidence:** 4

**Review:**

This paper describes a method for dealing with large action spaces where a partial model of the environment is learned which predicts the likelihood that a certain action will be 'valid' for a given state.  This model is learned in parallel to the agent's learning, and is used to prune the action space to make it more tractable.   If I understand correctly, the policy network continues to provide value predictions for the full set of N actions, but only a subset will be considered during the argmax calculation.  Therefore, this approach still scales linearly with N which is a bit disappointing, however it learns a sort of 'certainty' estimator that avoids spurious action predictions from the policy network from harming the final action choice (I am gleaming this from the last sentence citing Hester et al.).

I would however appreciate it if you make the action selection process slightly more clear in an eventual final submission, as well as detail the O(.) complexity of your algorithm, as I am not 100% sure I understood the exact way in which everything is put together.

Overall I think the idea of pruning the action set makes a lot of sense, both in large action spaces, but also in situations where experience coverage is not uniform over the action space.  I encourage more work in this direction.

---

### Decision · Program_Chairs · 2018-03-20
**ICLR 2018 Workshop Acceptance Decision**

**Decision:**

Accept

**Comment:**

Congratulations, your paper was accepted to the ICLR workshop.